# Evolution of Shock-Induced Pressure in Laser Bioprinting

**Evgenii Mareev** [1,2]📷, **Nikita Minaev** [1]📷, **Vyacheslav Zhigarkov** [1] **and Vladimir Yusupov** [1,*]📷

1   Federal Scientific Research Centre "Crystallography and Photonics", Institute of Photon Technologies, Russian Academy of Sciences, Pionerskaya St. 2, Troitsk, 108840 Moscow, Russia; mareev.evgeniy@physics.msu.ru (E.M.); minaevn@gmail.com (N.M.); vzhigarkov@gmail.com (V.Z.)
2   Faculty of Physics, M.V. Lomonosov Moscow State University, 119991 Moscow, Russia
*   Correspondence: iouss@yandex.ru

**Abstract:** Laser bioprinting with gel microdroplets that contain living cells is a promising method for use in microbiology, biotechnology, and medicine. Laser engineering of microbial systems (LEMS) technology by laser-induced forward transfer (LIFT) is highly effective in isolating difficult-to-cultivate and uncultured microorganisms, which are essential for modern bioscience. In LEMS the transfer of a microdroplet of a gel substrate containing living cell occurs due to the rapid heating under the tight focusing of a nanosecond infrared laser pulse onto thin metal film with the substrate layer. During laser transfer, living organisms are affected by temperature and pressure jumps, high dynamic loads, and several others. The study of these factors' role is important both for improving laser printing technology itself and from a purely theoretical point of view in relation to understanding the mechanisms of LEMS action. This article presents the results of an experimental study of bubbles, gel jets, and shock waves arising in liquid media during nanosecond laser heating of a Ti film obtained using time-resolving shadow microscopy. Estimates of the pressure jumps experienced by microorganisms in the process of laser transfer are performed: in the operating range of laser energies for bioprinting LEMS technology, pressure jumps near the absorbing film of the donor plate is about 30 MPa. The efficiency of laser pulse energy conversion to mechanical post-effects is about 10%. The estimates obtained are of great importance for microbiology, biotechnology, and medicine, particularly for improving the technologies related to laser bioprinting and the laser engineering of microbial systems.

**Keywords:** LIFT; shock wave; laser bioprinting; LEMS

## 1. Introduction

Laser-induced forward transfer (LIFT) is based on the transfer of a microscopic amount of matter because of laser pulsed heating. It is a promising alternative to traditional printing methods [1]. Currently, laser printing technologies are actively used in various fields, such as medicine and biotechnology, for printing with cell and microbial cultures to create artificial tissue-engineered and cell-engineered structures [2]. In these cases, the term laser bioprinting is widely used [3].

A promising modification of the laser printing method is laser engineering of microbial systems (LEMS) technology. It has already been proven that LEMS will make it possible to separate symbionts [4] and isolate individual microorganisms that are difficult to cultivate or not cultivated at all by standard methods [5–7]. It is important that this process is necessary not only for solving the urgent problem of creating the demanded "Noah's Ark"—the World Bank of Microorganisms [8]—but also for the production of biologically active substances [9] and the synthesis of new antibiotics [10].

In laser bioprinting or LEMS technologies, a hydrogel layer with cells or microorganisms is applied to a transparent donor plate with a nanometer absorbing layer [11–13]. A result of the impact of an infrared nanosecond laser pulse with a wavelength of 1064 nm on such a system is that the local area of the absorbing layer and the nearby thin layer of gel

are abruptly heated to temperatures exceeding supercritical values for water. Such heating leads to the explosive appearance of a rapidly expanding vapor–gas bubble, at the top of which a thin jet of gel is formed, from which a microdroplet with cells or microorganisms is then separated [14].

It is clear that with this laser method of transfer, microorganisms will be exposed to a number of physical factors—abrupt jumps in temperature and pressure near the surface of a metal film, high dynamic loads associated with the initial acceleration and subsequent "landing" of microdroplets, etc. [6,15]. On the one hand, such physical factors can negatively affect these living systems, leading to the suppression of their metabolism, partial destruction, and even death. On the other hand, some of them, under certain parameters, can act as triggers that stimulate the metabolism of microorganisms and ultimately lead to the positive results demonstrated by LEMS technology. Therefore, the experimental evaluation of various physical factors accompanying laser transfer is not only of purely theoretical interest but also of practical importance.

Recently, it was shown in [15] that, in the operating range of laser pulse energies (15–35 μJ with a laser spot diameter of 30 μm), when there is a stable transfer of hydrogel microdroplets with microorganisms, pressure jumps near the Au absorbing film pressure reach up to 150 bar. Using the results of measurements of acoustic waves in the far zone (when Ti and Cr absorbent films were used) and the literature data, it was found [16] that shock waves can appear at the upper edge of this energy range, and pressure jumps in the hydrogel layer with microorganisms can reach 5–10 kbar. The study of the shock waves arising near the absorbing metal layer can be effectively carried out using a time-resolving shadow microscope. This method makes it possible to study the dynamic processes associated with the generation and cavitation collapse of microbubbles [17], the appearance of jets [18], and shock waves in greater detail [19]. The purpose of this article was to study the parameters of dynamic processes in bioprinting and LEMS technologies, and to estimate the pressure jumps that occur in a liquid layer near an absorbing film during laser printing using time-resolved shadow microscope photography.

## 2. Materials and Methods

The schematic diagram of a setup for studying dynamic processes using time-resolving shadow microscopy is shown in Figure 1.

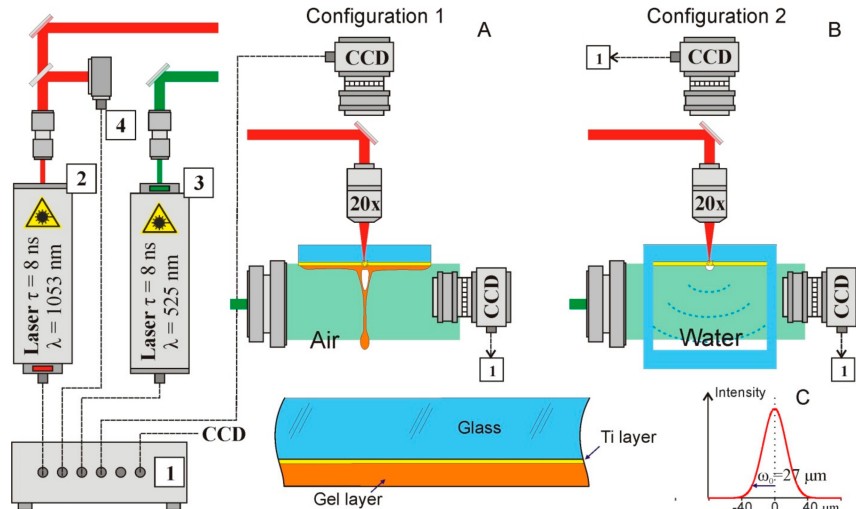

**Figure 1.** Schematic diagram of a setup for studying dynamic processes using time-resolving shadow microscopy using a donor glass plate with a thin Ti layer in two configurations: (**A**)—in LEMS geometry with a gel layer applied to the donor plate (enlarged image below) and (**B**)—in a glass mini-cell with water. (**C**)—intensity distribution in the laser spot. 1—real-time controller; 2—IR nanosecond laser; 3—visible nanosecond laser; 4—photodetector.

We used the technique of time-resolved shadow photography in the experiments. The first IR nanosecond pulse was tightly focused (NA = 0.2, focal length 2.5 cm) onto a thin metal film through a glass substrate. Shadow photographs were recorded using a 1.3 MP CMOS gigabit ethernet monochrome digital camera HT-GE134GM-TPO-CL (Mindvision, China); uniform illumination in the area was produced by transmitting a visible probe nanosecond pulse through a diffusion plate. The image was focused onto the CCD matrix by the same objective used for focusing the IR laser pulse; in this case, a spatial resolution of about 2.5 $\mu$m per pixel was achieved.

We used two laser sources Tech-1053 and Tech-527 (LaserExport, Moscow, Russia) with a nanosecond pulse duration ($\tau$ = 8 ns FWHM). Infrared ($\lambda$ = 1053 nm) laser impulse acts as a pump and a visible laser impulse ($\lambda$ = 527 nm) acts as a probe. The energy of the infrared impulse in experiments was up to 500 $\mu$J and the energy of the visible pulse was 275 $\mu$J. The repetition rate of the laser impulses was 2 Hz. To achieve the temporal resolution, we varied the delay between these impulses. For this purpose, we used an electronic delay system. We implemented this circuit based on FPGA and analog electronics. The main time-delay controller is the CompactRIO (National Instruments, Ostin, TX, USA). At the FPGA level, 2 trigger pulses were generated with a frequency of up to 1 MHz and a minimum step between pulses of 25 nanoseconds. The controller also runs a real-time system that communicates with a PC via a LAN and transfers data to the FPGA. Delayed fine tuning is conducted on a self-made analog circuitry that achieves 250 ps steps with a maximum latency of 64 ns. The jitter of the electronic part of the circuit (measured with a 1 GHz oscilloscope) is less than 1 ns. However, the jitter between receiving the trigger and directly generating the laser pulse is in the order of 250 ns. To compensate for this jitter, many (more than 50) measurements were performed for each time delay, the results of which were averaged. Thus, by varying the time delay between the pump and probe impulses, we observed the evolution of shock waves and jets. The entire system was controlled using specially developed software on LabVIEW, which provided control of the repetition rate of laser impulses and a variation in the delay between the pump and probe laser pulses with a step of 250 ps. The software also performs recording from the camera and synchronization with the rest of the system. The experiment was fully automated, allowing us to record the complete evolution of the system.

The experiments were carried out with AG00000112E (26 $\times$ 76 $\times$ 1 mm) donor glass plates (Menzel, Thermo Fisher Scientific, Berlin, Germany), on which a 50 nm-thick Ti layer was deposited using magnetron sputtering.

The experiments were carried out in two setup configurations (Figure 1). In the first series of experiments (Configuration 1 in Figure 1A), dynamic processes were studied in the usual for LEMS configuration. The radiation of a pulsed laser with $\lambda$ = 1053 nm was focused on a Ti-absorbing film of a donor glass plate, on which a hydrogel layer was previously deposited (enlarged image in the lower part of Figure 1A). The rapid heating of the metal film [13] leads to the generation of a rapidly expanding vapor–gas bubble in the region of laser impact. At the top of this bubble, after some time, a thin jet of hydrogel was formed. All these processes were recorded with the required time resolution using time-resolving shadow micro photography. During experiments performed in Configuration 2 we did not register shock waves. That is the result of the geometry of the experiment. The jets were generated in the center of the glass plate 26 $\times$ 76 $\times$ 1. The gel efficiently scatters the probe pulse propagated through it, in a result at the shadow photographs it was represented as a dark area, where no details could be observed.

In the most of experiments a gel based on hyaluronic acid (2% in water) was used. The viscosity of the gel, determined using a *micro*VISC viscometer (RheoSense, San Ramon, CA, United States), was 15.5 $\pm$ 0.1 mRa·s. However, in the biological experiments living cells in the distilled water were added to the gel. Thereby, the gel concentration, and viscosity, decreased. To illustrate this process, we also used the gel with 1.5% hyaluronic acid. The detailed description of the viscosity role during LEMS could be found in [14].

Immediately before these experiments, a $200 \pm 30$-μm-thick gel layer was applied to the donor plate from the side of the Ti layer. For this, the plate was placed horizontally on a motorized slide, and a drop of gel was smeared with a knife blade installed with the required clearance above the surface.

In the second series of experiments, dynamic processes were studied near an absorbing film, which was in contact with water (Configuration 2 in Figure 1B). In these experiments, we used a glass cell, one wall of which was made of a standard donor plate with a 50 nm Ti layer deposited. Time-resolving shadow microscope recording was used to record the shock waves and vapor–gas bubbles formed near the donor plate.

The laser pulse energy was measured by an S310C meter (Thorlabs). The results were presented as mean and standard deviation.

To estimate the pressure jumps arising in the region of laser impact, we used a quasi-empirical equation that unambiguously related the magnitudes of these jumps to the velocity of the shock wave front in water [20,21]:

$$p_s = s_1 \rho_0 u_s (10^{(u_s - c_0)/s_2} - 1). \tag{1}$$

where $u_s(r, t)$ is the shock front velocity; $p_s(r,t)$ is the pressure of the shock wave; $\rho_0$ is the density of water; $c_1$ and $c_2$ are empirical constants: $c_1 = 5190$ m/s and $c_2 = 25,306$ m/s; $c_0$ is the sound speed; and $r$ is the radius.

The shock wave energy, following [22], was estimated as the integral of the shock wave energy losses during its propagation:

$$E = \int_{r_2}^{r_1} 4\pi r^2 \rho(r) \Delta\varepsilon(r) dr, \tag{2}$$

where $\Delta\varepsilon$ is defined as:

$$\Delta\varepsilon(r) = \frac{1}{2}\left(\frac{1}{\rho_0} - \frac{1}{\rho(r)}\right) p(r). \tag{3}$$

The shock wave energy was calculated numerically by integrating Equation (2) in Python.

In addition, the energy of the cavitation bubble was estimated from the obtained shadow images. For this, the expression connecting the energy stored in the cavitation bubble $E_b$ with its maximum radius $R_{max}$ was used:

$$E_b = \frac{4\pi}{3}(p_{stat} - p_v)R_{max}^3, \tag{4}$$

where $p_v$ is the pressure of saturated vapor in the bubble; $p_{stat}$ is the pressure of the environment. For water, $p_{stat} = 100$ kPa (1 bar) and $p_v = 2.33$ kPa at 20 °C. To simulate the dynamics of the cavitation bubble, the Rayleigh relation was used:

$$\rho R \ddot{R} + \frac{3}{2}\rho \dot{R}^2 = p_i - p_e. \tag{5}$$

where $p_i$ is the pressure inside; $p_e$ is the pressure outside the bubble, that is equal to atmospheric pressure 0.1 MPa; $R$ is the cavitation bubble radius; and $\rho$ is the density of the medium. Additionally, we assumed that during collapse of the cavitation bubble 50% of the energy is lost, and this model has previously shown excellent coincidence with experimental data [23]. The Equation (5) was simulated using Python script. We varied the initial pressure (t = 0, R = 0), until reaching the obtained in the experiment maximal bubble size.

## 3. Results

In the first series of experiments, we studied the dynamic processes under the laser pulse impact on a donor plate with a layer of hydrogel deposited on its surface in a

configuration usual for LEMS technology (Configuration 1 in Figure 1A). The impact of a nanosecond laser pulse on the absorbing Ti layer of the donor plate led, in this case, to the generation of a vapor–gas bubble and a thin gel jet (Figure 2). The laser-induced plasma is not presented on Figures 2 and 3 because it is located inside the donor plate below the gel layer, which is used as a border of the figures.

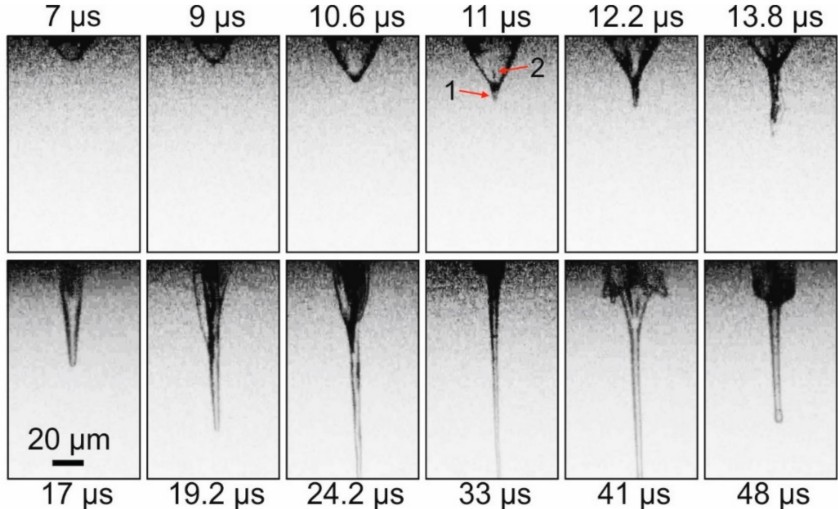

**Figure 2.** A sequence of frames obtained using time-resolved shadow microscopy during the generation of a vapor–gas bubble and a jet of gel (1.5% hyaluronic acid—see Methods) under the impact of a laser pulse with a duration of $\tau$ = 8 ns and an energy of E = 32 $\mu$J on the absorbing Ti layer in Configuration 1 (Figure 1A). Each frame shows the time after exposure to the laser pulse. The arrow (1) points to the jet, arrow (2) points to the counter-jet.

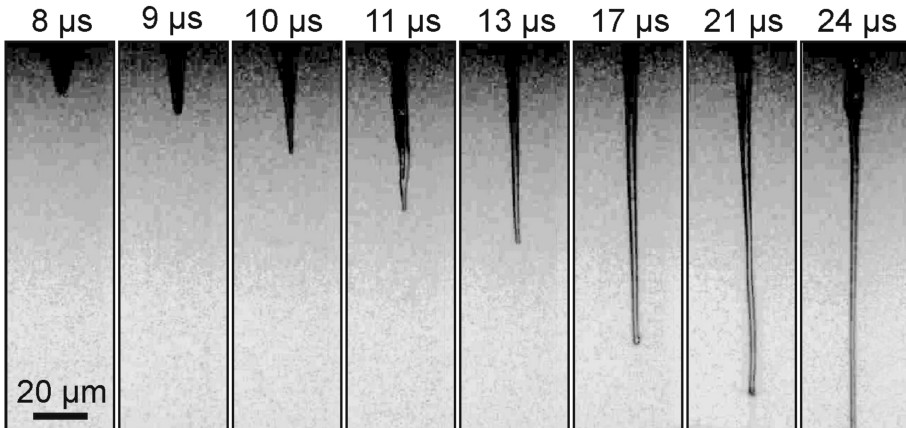

**Figure 3.** A sequence of frames obtained using time-resolved shadow microscopy during the generation of a vapor–gas bubble and a jet of gel (2% hyaluronic acid—see Methods) under the laser pulse impact with a duration of $\tau$ = 8 ns and an energy of E = 30 $\mu$J on the absorbing Ti layer in Configuration 1 (Figure 1A). Each frame shows the time after exposure to the laser pulse.

Figure 2 demonstrates that, initially, a rapidly growing vapor–gas bubble forms in the gel layer. At the top of this bubble, approximately 11 $\mu$s after the laser pulse, a thin jet (arrow 1 on Figure 2) of gel directed from the donor plate begins to form. At the same time, at the top of the bubble, a counter-jet directed towards the donor plate and spreading inside the bubble begins to form (arrow 2 on Figure 2). Note that, by this time, the bubble reaches its maximum size of ~34 $\mu$m. From this moment, the gel jet begins to rapidly increase in size, and the vapor–gas bubble begins to collapse gradually. By 33 $\mu$s after the laser impact, the vapor–gas bubble is no longer visible, and only a thin jet of gel is present in the frame,

the lower part of which is already outside the frame. Subsequently, the counter-jet and the bubble begin to expand away from the surface of the donor plate (see frame 41 μs), wherein the bubble surface does not look as smooth as it does during its initial frames of expansion process, and it is strongly perturbed. In the 48 μs frame, one can see how, at this moment, the remainder of the gel jet returns to the donor plate. The maximum recorded velocities for E = 32 μJ were the following: for a bubble during its expansion—9.3 ± 1.2 m/s; for a gel jet—8.6 ± 0.6 m/s.

Figure 3 shows a sequence of shadow images obtained by generating a gel jet under the same conditions as in Figure 2, but with a slightly lower laser pulse energy (Configuration 1 in Figure 1A).

From Figure 3, it is clearly seen that, when a laser pulse is focused into the absorbing layer of a donor plate for gel with a lower viscosity than in the previous example, the shadow images differ significantly from those shown in Figure 2. A thin jet of gel is also clearly visible on these frames, but the vapor–gas bubble is practically invisible. In this case, the maximum recorded velocity for the gel jet is lower than in the case of gel with 1.5% viscosity and amounts to 7.6 ± 0.5 m/s.

In the second series of experiments, we studied the dynamic processes under the laser pulses impact on a donor plate placed in a microcuvette with distilled water (Configuration 2 in Figure 1B). The distilled water was chosen as a sample due to the high amount of experimental data and as a good substitute for biologic tissues [24]. The pressure achieved during laser impact onto donor plate would not practically depend on the sample (Gel layer) thereby by retrieving the pressure in water we can estimate the achieved in gel pressure. Figure 4 shows typical shadow micro photography obtained in this configuration.

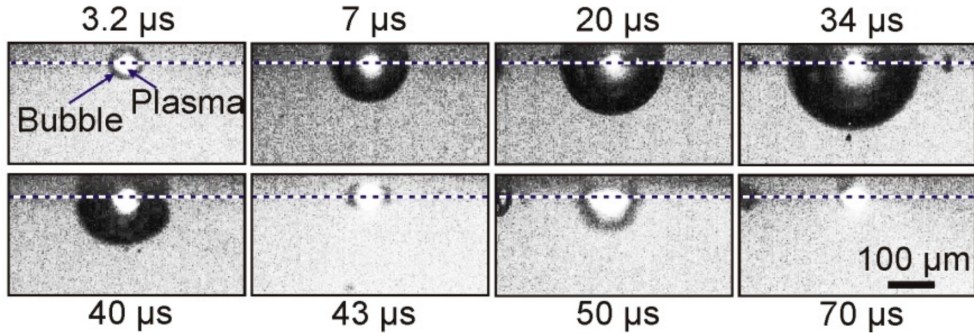

**Figure 4.** A sequence of frames obtained using time-resolved shadow microscopy during the generation of a vapor–gas bubble in water under the impact of a laser pulse with a duration of τ = 8 ns and an energy of E = 68 μJ on the absorbing Ti layer in Configuration 2 (Figure 1B). Each frame shows the time after exposure to the laser pulse. The dotted line shows the position of the Ti film on the lower surface of the glass plate.

As a result of the absorption of a laser pulse with an energy of E = 68 μJ in a Ti film of a donor plate, brightly glowing plasma, and a growing vapor–gas bubble is formed in the nearby region of laser impact in a microcuvette with water. Approximately 34 μs after laser exposure, this bubble reaches its maximum size, with a radius of ~150 μm, and then begins to collapse. Further frames show that, after bubble collapse, its pulsations occur, which are accompanied by fluctuations in size.

Figure 5 presents the dependences of the diameter of the cavitation bubble in water on time for three different energies of the laser pulse. Red curves modeling the dynamics of cavitation bubbles obtained using relation (5) are plotted in the same figures. The initial pressure in bubble was the fitting parameter (until the calculated maximal radius will be equal to the obtained in the experiment). The initial pressure inside the bubble is lower than the shock wave pressure and lies in range 40–100 MPa.

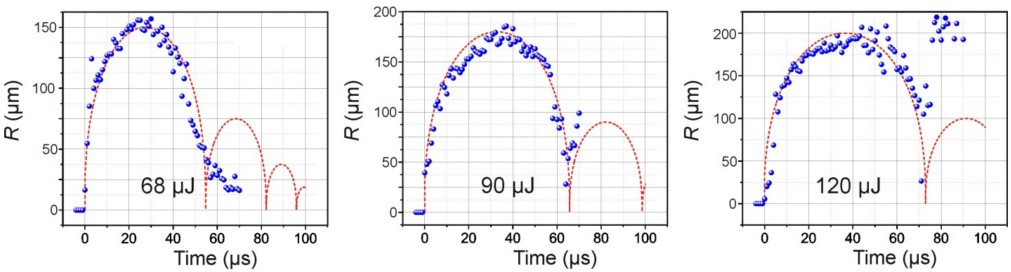

**Figure 5.** Time dependence of the diameter of a cavitation bubble in water at various laser pulse energies. Individual points were obtained by processing sequences of frames of time-resolving shadow microscope under the impact of laser pulses with $\tau$ = 8 ns on the absorbing Ti layer in Configuration 2 (Figure 1B). The red curves show the dependences of the bubble radii $R(t)$ calculated by Formula (5) at different values of the initial pressure 41, 49 and 59 MPa respectively.

As expected, an increase in the laser pulse energy leads to an increase in the maximum size of the formed bubble and the period of its pulsations (Figure 5). Moreover, its behavior also changes significantly after the first collapse. Therefore, at E = 68 µJ, the amplitude of the second oscillation of the bubble does not exceed 35 µm; at E = 90 µJ, it reaches 100 µm, and at E = 120 µJ, its rebound occurs at ~220 µm.

The results shown in Figures 4 and 5 reflect the existence of pronounced dynamic processes in a glass cell with water in the interval 1–100 µs after a laser pulse impact. Optical observation of these processes in a cell with water (Configuration 2 in Figure 1B) in the range of 0–1 µs made it possible to detect the generation of shock waves caused by the rapid heating of the Ti film (Figure 6).

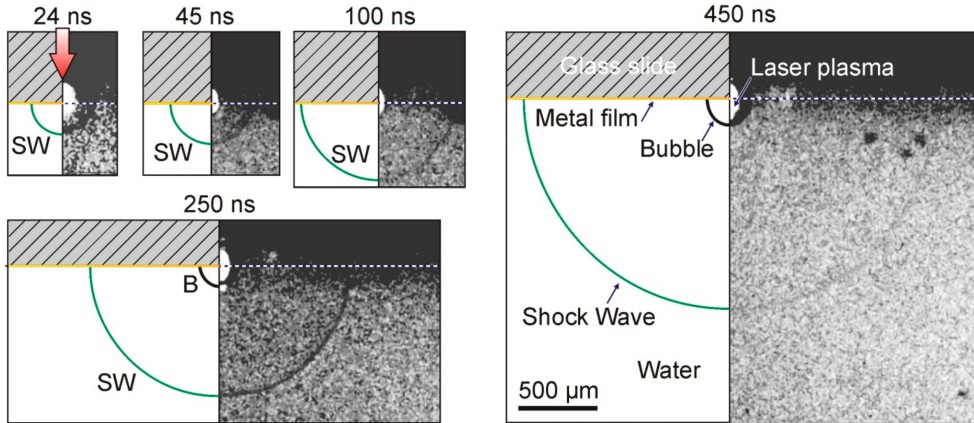

**Figure 6.** A sequence of frames obtained using time-resolving shadow microscopy during the generation and propagation of a shock wave (SW) in water under the impact of a laser pulse with a duration of $\tau$ = 8 ns and an energy of E = 68 µJ on the absorbing Ti layer in Configuration 2 (Figure 1B). Each frame shows the time after exposure to the laser pulse. In the last two frames, an expanding vapor–gas bubble (B) is recorded near the area of laser exposure. On the left side of each frame, a section of a donor plate with a glass slide, an absorbing metal layer, and the fronts of an expanding bubble and a shock wave are schematically shown. The red arrow in the first frame shows the direction of the laser pulse.

As can be seen from Figure 6, the impact of a laser pulse with an energy of E = 68 µJ on the absorbing Ti layer led to the generation of a plasma cloud, a shock wave, and an expanding vapor–gas bubble in the water. The shock wave is visible in the shadow picture only 24 ns after the laser pulse. It is also clearly visible in all presented frames up to $t$ = 450 ns. As for the vapor–gas bubble, it is recorded for the first time with a delay in the order of 250 ns.

Figure 7A shows the experimental points reflecting the time variation in the radius of the shock wave, as measured from shadow photographs at a laser pulse energy of $E$ = 68 μJ.

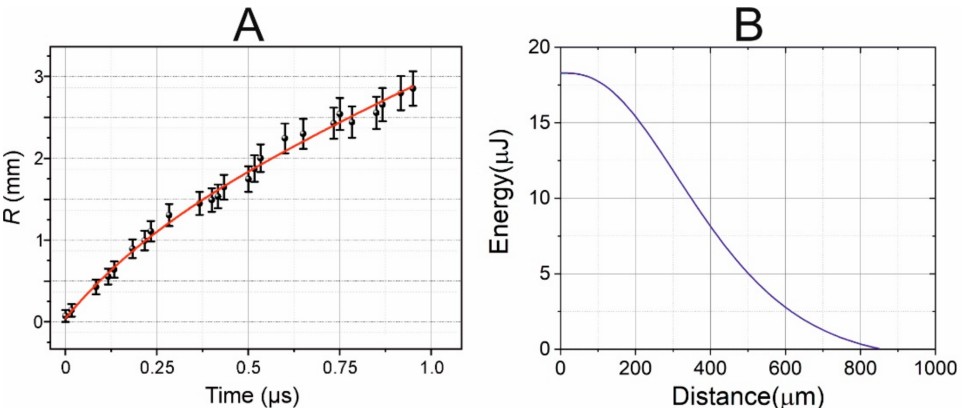

**Figure 7.** (**A**) Time variation in the shock wave radius in water under the impact of a laser pulse with a duration of τ = 8 ns and an energy of $E$ = 68 μJ on the absorbing Ti layer in Configuration 2 (Figure 1B). The red curve shows the trend for the exponentially decaying shock wave. (**B**) Dependence of shock wave energy on the distance from its center.

The data obtained (Figure 7) show that, as expected, the shock wave velocity is maximal in the beginning (at $t$ = 0), and gradually decreases with time while it approaches the speed of sound in water. To find the correct form of this trend, we assume that the shock wave velocity $V_{SW}$ decays exponentially with time $t$: $V_{SW} = c_0 + \Delta \cdot \exp(-a \cdot t)$; where $c_0$ is the speed of sound in water, $\Delta$ is the excess over the speed of sound, and $a$ is the attenuation coefficient. After integrating this dependence and taking into account the results obtained (Figure 7), we can find the following equation for the trend $R$(mm) = 1.48·$t$(μs) + 1.596·(1 − exp(−2.45·$t$(μs))), as shown in Figure 7 (red curve).

From the shadow photographs, we retrieved the dependence of the shock wave radius on time. From this dependence, assuming the exponential decay of the shock wave, we calculated the shock wave velocity. Using Equation (1), we calculated the pressure for different energies of the laser pulse. The maximal achieved pressure is about 1.75 GPa and the minimal registered is about 200 MPa (for 68 μJ laser pulse). The shock wave pressure has a $1.5 \cdot 10^{-4} \cdot E^{1.5}$ dependence on laser pulse energy, thereby for used in the LEMS process laser pulse energies (32 μJ) the shock wave pressure is about 27 MPa. The pressures are lower than the presented thresholds of cell damages under shock impact [25,26].

From Equations (2) and (3) we retrieved the dependence of the shock wave energy on the traveled distance. Figure 7B shows that the shock wave decays to the acoustic at distances about 850 μm. Furthermore, at distances less than 200 μm (thickness of the gel layer) about 25% of shock wave energy is dissipated, and it is in this area that the destructive effect on cells will be maximal. The performing measurements at the far zone (>1 mm) from the shock wave source (for example, with hydrophone) could not give the direct measurements of the shock wave pressure and must apply complex calibration procedure, including several assumptions, because information about shock-wave high-harmonics is lost during propagation [16].

Moreover, using Equations (2) and (4), we also calculated the energy of the shock wave and cavitation bubble. Figure 8B shows the dependence of the energy of a shock wave and a cavitation bubble arising in water, as calculated by Formulas (2) and (3). As such, according to our estimates, the efficiency of conversion of the laser pulse energy to mechanical post-effects is about 10%, and the conversion decreases at higher laser pulse energies.

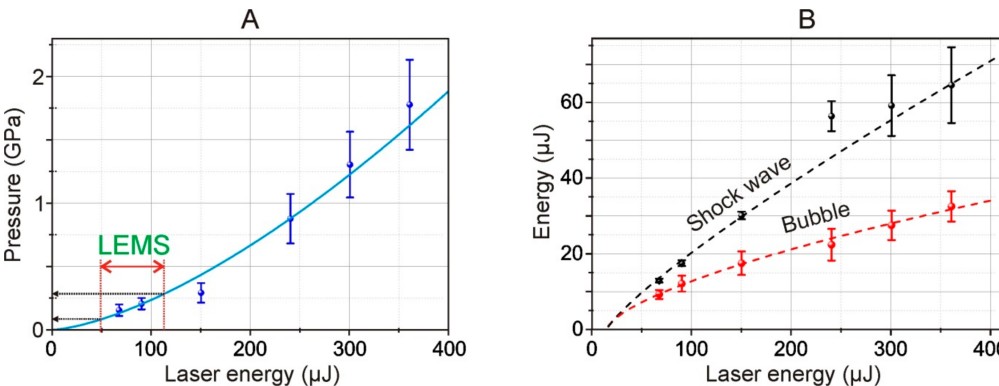

**Figure 8.** Dependencies of the shock wave pressure (**A**) and the energies (**B**) of a shock wave and a cavitation bubble arising in water from laser energies when a laser pulse duration of $\tau = 8$ ns is applied to an absorbing Ti layer in Configuration 2 (Figure 1B). The blue line in Figure 8A shows trend $1.5 \cdot 10^{-4} \cdot E^{1.5}$. The dotted lines in Figure 8B show the corresponding trends $0.4 \cdot (E-20)^{0.83}$ for the bubble and $0.59 \cdot (E-20)^{0.65}$ for the shock wave. Figure 8A highlights the area corresponding to the operating range of fluences for LEMS technology. The vertical dotted arrows limit the range of pressure jumps that can be generated in LEMS technologies.

The decrease of the efficiency of the laser induced post effects is caused by the increase of the area of the laser impact whereas the energy conserved in the electron plasma remains virtually unchanged. Thereby the deposited energy density drops [27]. From the viewpoint of practical use, the most preferred mode of LIFT is working at the laser energies close to the threshold energies of shock wave generation. In such a regime the achieved pressures are high enough for shock wave generation, and the impact area is minimal.

## 4. Discussion

In a real laser printing on a donor plate (a glass plate with a thin absorbing metal layer), a layer of a gel substrate is applied. The gel may contain biomolecules [28], living cells [29], or microorganisms [6,7]. The impact of tightly focused ns laser pulses leads to the local sharp heating of the thin metal layer and the generation of a rapidly expanding cavitation bubble in the gel layer [14]. This leads to the appearance of a thin jet (as in Figures 2 and 3). Further, a small volume of gel that contains a small number of living cells or microorganisms is transferred to the receiving media.

The impact of a laser pulse with E = 32 μJ on the absorbing Ti layer of the donor plate with a gel layer in Configuration 1 (Figure 1) led to the appearance of a bubble rapidly expanding with a velocity of 9.3 ± 1.2 m/s and the formation of a thin jet of gel, the velocity of which was 8.6 ± 0.6 m/s (Figure 2). A decrease in the energy to E = 30 μJ led to the bubble standing out less distinctly in the shadow micrographs, and the maximum velocity of the gel jet decreased to 7.6 ± 0.5 m/s (Figure 3). Experiments carried out in a cuvette with water in Configuration 2 (Figure 1) showed that a short-lived vapor–gas bubble appeared near the absorbing layer as a result of nanosecond laser impact (Figure 4). In this case, the maximum size of this bubble and the period of its pulsations increase with an increase in the energy of the laser pulse (Figure 5); these increases are well described by Formula (5).

Time-resolving shadow micro-photography in Configuration 2 (Figure 1) also revealed the occurrence of shock waves propagating in water (Figure 6). Their speed is maximal at the initial moment of time, and it gradually decreases during wave propagation, approaching the speed of sound in water (Figure 7). Using quasi-empirical Equation (1), a curve was obtained that reflects the dependence of the shock wave pressure on the laser energy (Figure 8A). In addition, we revealed the dependences of the energy of a shock wave and a cavitation bubble arising in water on the energy of a laser pulse using Formulas (2) and (3) (Figure 8B). These dependences made it possible to conclude that the conversion of the

laser pulse energy into mechanical post-effects is about 10%, and the conversion decreases for higher laser pulse energies.

In this work, we attempted to assess what pressure surges are experienced by living systems in the process of laser bioprinting. To use the results obtained in this article (presented in Figures 4–8) to answer this question, it is necessary to determine the range of laser energies of our system (Figure 1), which would correspond to the range of operating energies of the systems used for bioprinting. In the case of LEMS technology, the operating energy range, in which a controlled and stable transfer of gel microdroplets occurs, is 15–35 μJ (with a laser spot diameter of 30 μm) [15].

Ultimately, the processes occurring during laser printing are determined by the speed and maximum heating temperature of the metal film. Note that, in the LEMS operating energy range, the maximum temperatures exceed the melting point of metal [30] and lead to partial destruction of the metal film [31]. Therefore, the amplitudes of the temperature jumps $\Delta T$ in the region of the laser pulsed impact will be proportional to the laser fluences $F$ minus a certain value C, which is determined by the energy spent on phase transitions. Assuming that the values of C in our case and in LEMS bioprinting technology are equal, we find that the jumps $\Delta T$ are proportional to $F = 2E/(\pi \cdot w_0^2)$, where $E$ is the laser pulse energy and $w_0$ is the radius of the waist in the laser spot. Thus, the pulse energy of our system $E$ (Figure 1) which, according to the processes taking place, will correspond to the energy of the laser pulse in the LEMS technology $E_{LEMS}$, is:

$$E = E_{LEMS}\left(\frac{w_0}{w_{0LEMS}}\right)^2,\qquad(6)$$

where $w_{0LEMS}$ = 15 μm—radius of the waist in LEMS technology [15].

In Figure 8A, the double red arrow marks the 49–113 μJ region which, using recalculation according to (6), corresponds to the operating energy range of 15–35 μJ for LEMS gel microdroplet transfer technology [15]. From the trend shown in Figure 8A, it follows that such energies correspond to pressure jumps in the range of 80–280 MPa (800 bar–2.8 kbar) occurring near the metal film in the liquid in Configuration 2 (Figure 1). Note that this maximum value (2.8 kbar) is close but somewhat lower than the estimated value given in [16] (5 kbar for the Ti film).

Shock waves with a supersonic high amplitude pressure jumps and a short pulse duration can affect microbiological systems. First, let us estimate the magnitude of the density jump during the passage of the shock wave. It can be assessed from the modified Tait equation of state for water [32,33]:

$$P = (P_0 + B) \cdot (\rho/\rho_0)^\gamma - B,\qquad(7)$$

where the constant $B$ = 304 MPa; $\gamma$ = 7.15; $P$ and $\rho$ are the pressure and density in shock wave; and $P_0$ and $\rho_0$ are the atmospheric pressure and reference density, respectively. From (7), for the maximum pressure in the shock wave $P$ = 280 MPa, we obtain $\rho/\rho_0 \approx 1.1$. We suggest that such a seemingly insignificant increase in density for pure water (by 10%) can have a significant effect on more compressible microorganisms which, as is known, have numerous microcavities inside them.

In the works by A.G. Doukas et al. and M. Shmidt et al. [34,35] the authors have focused on correlating cell damage to the peak pressure and the stress gradient. In a study performed by V.E. Gusev and colleagues [36], using cancer cells as an example, it was demonstrated that exposure to shock waves with a pressure amplitude of ~40 MPa can lead to the death of up to 80% of cells. On the other hand, the same article shows that the impulse (the integral of pressure over time), rather than peak pressure, is the governing shock wave parameter for viability.

In a study done by M. Schmidt and colleagues [34], it was concluded that the rise time in shock waves plays a significant role in biological effects in vitro. The arising pressure gradient on the microbiological object depends on this value. In LEMS technology, the

duration of the laser pulse is 8 ns, so the duration of the generated acoustic pulse will be approximately the same [37]. In this case, the acoustic wavelength will be ~ 10 microns, which roughly corresponds to the size of the cells and microorganisms being transferred (erythrocytes ~7 microns, *E. coli* 2–6 microns, stem cells ~20 microns). In a shock wave, due to the very small length of the front and high-pressure jumps, the generated pressure gradients can be significantly large. For example, in [33], it was shown that, with a geometry similar to our experiment and at a laser pulse duration of 7 ns, the pressure rise time at the shock front turned out to be <1 ns, which, for a propagation velocity of 2 km/s, gives a characteristic distance of <2 μm.

From the point of view of laser bioprinting technology, the question of the effect of the generated shock waves on microbiological objects located in the gel layer of the donor plate is of interest. In this case, we should consider that, in real bioprinting, the absorbing metal film does not border on water but on the substrate, which has a higher viscosity. The increased viscosity will increase the losses during the propagation of the shock wave; therefore, the shock waves formed near the absorbing film will decay much faster [38]. In this regard, we note that one of the advantages of LEMS technology is its use of a relatively thick gel layer (200–300 μm). This makes it possible to reduce the negative consequences of a number of physical factors acting on living systems in the process of laser transfer. In particular, microbiological objects located in the lower part of the "thick" gel layer of the donor plate will no longer receive a shock wave but an ordinary acoustic wave. Note that such waves can positively affect living systems due to the well-known effect of mechanobiology [39,40], leading to cell stimulation [40,41]. It should be noted that similar effects of the stimulation of microorganisms during their transfer using LEMS technology are known [5].

## 5. Conclusions

In this article, the parameters of dynamic processes in laser bioprinting by laser-induced forward transfer technologies were investigated using time-resolved shadow microscopy. We presented the experimental results of the study parameters of the process formation of bubbles and jets of gel during laser printing by gel microdroplets upon absorption of a laser pulse with $\lambda$ = 1053 nm and $\tau$ = 8 ns in a thin Ti layer of a donor plate. It was shown that shock waves with pressures of up to 280 MPa appeared near the absorbing film of the donor plate in the operating range of laser energies for bioprinting LEMS technology. However, from the obtained trends in the operating range of laser energies the shock wave pressures were about 30 MPa, which is lower than the thresholds of living cells destruction. It was shown that the conversion of the laser pulse energy to mechanical post-effects is about 10%, and the conversion decreases for higher laser pulse energies. The results obtained are important for tuning the parameters of the laser printing mode, which makes it possible to reduce the negative consequences of a number of physical factors acting on living systems in the process of laser transfer. The estimates obtained are of great importance for microbiology, biotechnology, and medicine, particularly in relation to improving the technologies of laser bioprinting by laser-induced forward transfer and the laser engineering of microbial systems.

**Author Contributions:** Conceptualization, V.Y. methodology, E.M., V.Z.; software, E.M.; validation, V.Y. and N.M.; formal analysis, V.Y., V.Z.; investigation, E.M., V.Y. and V.Z.; resources, N.M.; data curation, E.M.; writing—original draft preparation, V.Y.; writing—review and editing, V.Y., N.M. and E.M.; visualization, E.M. and V.Y.; supervision, V.Y.; project administration, N.M.; funding acquisition, N.M., E.M. and V.Y. All authors have read and agreed to the published version of the manuscript.

**Funding:** This work was supported by the grant from the Russian Science Foundation 20–14-00286 in terms of conducting an experiment, performing evaluations, studying the result obtained, formulating the main conclusion and improving bioprinting and LEMS technology; partly supported by the Ministry of Science and Higher Education within the framework of the State Assignment of the Federal Research Center "Crystallography and Photonics" of the Russian Academy of Sciences in terms of using the equipment of the Center for Collective Use by providing pulse laser equip-

ment; partial supported by the RFBR grant 19-32-60072 in terms of creating time-resolved shadow photography setup.

**Institutional Review Board Statement:** Not applicable.

**Informed Consent Statement:** Not applicable.

**Data Availability Statement:** The data presented in this study are available on request from the corresponding author. The data are not publicly available due to privacy.

**Conflicts of Interest:** The authors declare no conflict of interest.

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
