# Peer review of "Evolution of Shock-Induced Pressure in Laser Bioprinting"

_photonics, doi:10.3390/photonics8090374_

Round 1

Reviewer 1 Report

The authors present a study on laser-induced shockwave generation using an experimental configuration that has common features with the one used in laser-induced forward transfer (LIFT). They discuss potential implications of the findings in laser bio-printing of microbial systems. The study is interesting and follows a series of papers from the same group dealing with this research question.   

Major concerns:

Although the shockwave study is comprehensive and well presented, the overall work might be considered incremental considering https://doi.org/10.1016/j.optlastec.2020.106806 and https://doi.org/10.1070/QEL17081, where direct measurement of acoustic waves is presented using a similar experimental setting. I remain sceptical on the additional value of the conclusions presented here (i.e., analysis based on shockwave visualization). What do we learn new about the generated pressure waves (beyond visualization) compared to the previous approach used by the authors?  

The authors must address the fact that shockwave generation remains unnoticed in LIFT (i.e., donor + 200 um hydrogel) performed at the laser energy density used in Configuration 2. An explanation is required. Shock waves have been previously captured in both liquid and solid state LIFT.     

Minor points:

The title is inaccurate. There is non shockwave detection/study in a bioprinting setting.

There is an obvious dumping effect in bubble oscillation (Fig 5), but there is no relevant dumping term in equation 5. The constants and fitting parameters used for modeling must be provided.

Contrary to Configuration 2, plasma is not visible in Configuration 1. An explanation is required.   

The rational for presenting typical LIFT jets at 32 uJ (Fig 2) and 30 uJ (Fig 3) is not clear. One may argue that the difference lies within the typical uncertainty of the measurement.     

Reviewer 2 Report

In their work, Mareev et al. investigated the operating parameters of the dynamic processes in laser engineering of microbial systems (LEMS) technology by laser-induced forward transfer (LIFT). In addition, the authors estimated the pressure jumps that occur in a liquid layer near an absorbing film during laser printing. This study addresses one of the hot topics regarding LIFT bioprinting. The manuscript is well prepared, and their findings represent a major contribution to the improvement of laser bioprinting technology in general. I recommend its publication after the following comments are clarified:

  • Line 356-358: the authors obtained an almost halved maximum value of pressure jumps (2.8 kbar) compared to the estimated value (5kbar) from the group’s previous work (reference [16]). Please clarify this difference in more detail.
  • The obtained maximum pressure in the shock wave of 280 MPa is well above the reported critical pressure thresholds that cause cell damage ( ̴40 MPa and 80 MPa, respectively). Can the authors predict the extent to which exposure to such high pressure will affect living cells?
  • According to a systematic review by Antoshin et al. [ANTOSHIN, A. A., et al. LIFT-bioprinting, is it worth it?. Bioprinting, 2019, 15: e00052.], one of the potential problems associated with LIFT bioprinting may be laser evaporation of metal energy absorbing layer (EAL), which leads to the formation of nano- and microparticles. The latter can be potentially toxic to living cells after transfer to a collector substrate. Also, have the authors considered how shock wave treatment might affect the quality of deposition of a metal EAL?

Lastly, I have a few suggestions that are more “cosmetic repair”:

  • Better resolution of the shadow microscopy images (Figure 2, 3, and 4).
  • The authors could add spaces between the main text and the captions for better clarity in the manuscript.
  • Lines 371 and 376: name the referenced work, for example:

“In the work by Liao et al. [33]…”

“In a study done by Doukas and colleagues [31]…”

Round 2

Reviewer 1 Report

The authors have addressed my concerns and comments. Yet, two minor points should be addressed before publication. 

During experiments performed in Configuration 2 we did not register shock waves. That is the result of the geometry of the experiment. The jets were generated in the center of the glass plate 26x76x1. The gel efficiently scatters the probe pulse propagated through it, in a result at the shadow photographs it was represented as a dark area, where no details could be observed.

A note must be added in the manuscript stating this limitation. The question of why not visualizing directly using a LIFT setup may naturally arise to the readers.    

During fitting, we varied the maximal size of the bubble. It equals to 150, 175 and 200 m for Fig.5 a,b and c respectively.

It remains not clear to me. I assume you played with pe at t=0 or R at t=0 or both to hit a solution that will match the experimental radius. You need to clarify that in the manuscript and provide the required info (pe0, R0, other?) the reader would need to find the same solution when fitting those three sets of data.    

Author Response

First of all, we want to thank the Reviewers for their work. The all changes in the manuscript are presented in the red-line version of the manuscript. The answers for the issues are given below.

Point 1:

During experiments performed in Configuration 2 we did not register shock waves. That is the result of the geometry of the experiment. The jets were generated in the center of the glass plate 26x76x1. The gel efficiently scatters the probe pulse propagated through it, in a result at the shadow photographs it was represented as a dark area, where no details could be observed.

A note must be added in the manuscript stating this limitation. The question of why not visualizing directly using a LIFT setup may naturally arise to the readers.   

Answer:

We added this text fragment to the Methods section (Lines 126-130)

 Point 2:

During fitting, we varied the maximal size of the bubble. It equals to 150, 175 and 200 m for Fig.5 a,b and c respectively.

It remains not clear to me. I assume you played with pe at t=0 or R at t=0 or both to hit a solution that will match the experimental radius. You need to clarify that in the manuscript and provide the required info (pe0, R0, other?) the reader would need to find the same solution when fitting those three sets of data.

 Answer:

We completely agree with this remark. Indeed, R0 is the generalized radius of the bubble, used in the numerical simulation. Introducing this term would complicate the text, so we give the values of the initial pressures that demonstrates the best fitting. In the current version of the manuscript, we added the following text fragments (L.173-174, L244-248):

“The Eq. (5) was simulated using Python script. We varied the initial pressure (t=0, R=0), until reaching the obtained in the experiment maximal bubble size.”
“The initial pressure in bubble was the fitting parameter (until the calculated maximal radius will be equal to the obtained in the experiment). The initial pressure inside the bubble is lower than the shock wave pressure and lies in range 40 – 100 MPa.”